# Human genetic ancestry, *Mycobacterium tuberculosis* diversity, and tuberculosis disease severity in Dar es Salaam, Tanzania

Michaela Zwyer[1,2], Zhi Ming Xu[3,4], Amanda Ross[1,2], Jerry Hella[5], Mohamed Sasamalo[5], Maxime Rotival[6], Hellen Charles Hiza[1,2], Liliana K Rutaihwa[7], Sonia Borrell[1,2], Klaus Reither[1,2], Jacques Fellay[3,4,8], Damien Portevin[1,2], Lluis Quintana-Murci[6,9], Sebastien Gagneux[1,2]*, Daniela Brites[1,2]*

[1]Swiss Tropical and Public Health Institute, Allschwil, Switzerland; [2]University of Basel, Basel, Switzerland; [3]Swiss Institute of Bioinformatics, Lausanne, Switzerland; [4]School of Life Sciences, École Polytechnique Fédérale de Lausanne, Lausanne, Switzerland; [5]Department of Intervention and Clinical Trials, Ifakara Health Institute, Ifakara, United Republic of Tanzania; [6]Human Evolutionary Genetics Unit, Institut Pasteur, Université Paris Cité, Paris, France; [7]FIND, Foundation for Innovative New Diagnostics, Geneva, Switzerland; [8]Precision Medicine Unit, Lausanne University Hospital and University of Lausanne, Lausanne, Switzerland; [9]Chair of Human Genomics and Evolution, Collège de France, Paris, France

*For correspondence:
sebastien.gagneux@swisstph.
ch (SG);
d.brites@swisstph.ch (DB)

## eLife Assessment

This **valuable** observational study was conducted in Dar es Salaam, Tanzania, to investigate potential associations between genetic variation in *Mycobacterium tuberculosis* and human host vs. disease severity. The authors conclude that human genetic ancestry did not contribute to tuberculosis severity and the evidence supporting this is generally **convincing**. The findings have significance for the understanding of the influence of host/bacillary genetics on tuberculosis disease.

**Abstract** Infectious diseases have affected humanity for millennia and are among the strongest selective forces. Tuberculosis (TB) is an ancient disease, caused by the human-adapted members of the *Mycobacterium tuberculosis* complex (MTBC). The outcome of TB infection and disease is highly variable, and co-evolution between human populations and MTBC strains may account for some of this variability. Particular human genetic ancestries have been associated with higher susceptibility to TB, but sociodemographic aspects of the disease can confound such associations. Here, we studied 1000 TB patients from Dar es Salaam, Tanzania, together with their respective MTBC isolates, by combining human and bacterial genomics with clinical data. We found that the genetic background of the TB patient population was strongly influenced by migrations of Bantu-speaking populations from West Africa, which contrasts with the corresponding MTBC genotypes that were mainly introduced from outside Africa. These findings suggest a recent evolutionary history of co-existence between the human and MTBC populations in Dar es Salaam. We detected no evidence of an effect of human genetic ancestry, or MTBC phylogenetic diversity alone, nor their interaction, on TB disease severity. There was also no evidence of an association between human variation genome-wide and TB disease severity. Treatment-seeking, social, and environmental factors are likely to be the main determinants of disease severity at the point of care in this patient population.

## Introduction

Africa harbors the largest human genetic diversity worldwide (*Tishkoff and Verrelli, 2003*). This continent is also inhabited by numerous ethnic and linguistic groups (*Reed and Tishkoff, 2006*). While the long evolutionary history of modern humans in Africa and their large effective population sizes (*Garrigan et al., 2007*) explain this high genetic diversity, more recent migration events within and from outside Africa during the last 5000 years, as well as admixture between historically separated populations, have resulted in some degree of homogenization (*Patin et al., 2017*). Hence, African populations nowadays are often composed of a mixture of different genetic ancestries (*Bird et al., 2023*; *Pfennig et al., 2023*). One human migration that had a major influence on the population structure of present-day Africans is the so-called 'Bantu expansion', where Bantu-speaking (BS) groups migrated from central Western Africa southwards and eastwards, spreading farming technologies across sub-Saharan Africa, and admixing with local groups of hunter–gatherers and pastoralists (*Patin et al., 2017*; *Wang et al., 2020*). Recent human genetic studies have identified moderate population structuring among BS populations (*Semo et al., 2020*; *Sengupta et al., 2021*), yet admixture with local populations has impacted immune responses to infectious diseases (*Uren et al., 2017*). Since pathogens have been one of the strongest selective forces driving human evolution (*Fumagalli et al., 2011*), disease susceptibility and clinical outcomes can differ markedly between human populations. Genome-wide association studies (GWAS) have identified mutations altering the susceptibility to various infectious diseases (*Chapman and Hill, 2012*; *Newport and Finan, 2011*; *Lee et al., 2013*; *Akcay et al., 2018*), including tuberculosis (TB), which remains the main cause of human death due to a single infectious agent (*WHO, 2023*).

The bacteria that cause TB belong to the *Mycobacterium tuberculosis* complex (MTBC) and can be classified into ten human-adapted phylogenetic lineages: Lineage 1 (L1) to L10, plus several lineages adapted to different wild and domestic animal host species (*Zwyer et al., 2021*; *Guyeux et al., 2024*). While TB is a global problem, the human-adapted MTBC lineages differ in their geographical distribution. L2 and L4 occur worldwide, and other lineages are restricted to specific regions. Specifically, L1 and L3 mainly occur around the Indian Ocean (*Gagneux, 2018*), L5 and L6 are limited to West Africa (*de Jong et al., 2010*), and L7 only occurs in Ethiopia (*Wiens et al., 2018*). This phylogeographic population structure of the human-adapted MTBC led to the hypothesis that certain MTBC genotypes are locally adapted to their sympatric human populations (*Gagneux, 2018*). This hypothesis is supported by findings from cosmopolitan settings, where sympatric associations between the geographical origin of TB patients and their infecting MTBC strains were observed (*Reed et al., 2009*; *Gagneux et al., 2006*; *Baker et al., 2004*; *Gröschel et al., 2024*). Moreover, in immune-compromised individuals with HIV co-infection, these sympatric associations were disrupted (*Fenner et al., 2013*). The various MTBC genotypes also differ phenotypically (*Du et al., 2023*), regarding disease progression (*de Jong et al., 2008*), transmission (*Asare et al., 2018*; *Holt et al., 2018*), disease presentation (*Click et al., 2012*) and immune activation in vitro (*Manca et al., 2004*; *Arbués et al., 2025*) and ex vivo (*Coussens et al., 2013*).

Human genetic diversity has also been linked to differences in TB susceptibility. While, for example, *TYK2* has been associated with TB disease worldwide (*Boisson-Dupuis et al., 2018*), several human genetic loci were not consistently associated with TB in populations from different geographical regions but specific to certain populations (*Stein et al., 2017*; *Phelan et al., 2023*; *Bai et al., 2023*; *Png et al., 2012*; *Zheng et al., 2018*). Particular human genetic ancestries have also been found to play a role in the context of TB. People with higher proportions of native Peruvian genetic ancestry showed a higher risk of progressing to active TB (*Asgari et al., 2022*), and a higher proportion of San genetic ancestry was associated with an increased risk for TB among South African Coloured individuals (*Daya et al., 2014*). Differences in ethnicity have also been associated with different inflammatory profiles of TB patients at the time of presentation (*Coussens et al., 2013*). In addition to the effects of human and bacterial genetic diversity on TB, many social and environmental factors, as well as co-morbidities, are known drivers of TB. These include malnutrition (*Macallan, 1999*), HIV infection, diabetes (*Selwyn et al., 1989*; *Dooley and Chaisson, 2009*), poverty (*Spence et al., 1993*), smoking (*Bates et al., 2007*), and alcohol consumption (*Imtiaz et al., 2017*). While associations between TB and individual host, pathogen, or environmental factors have been found (*Stein et al., 2017*; *Phelan et al., 2023*; *Bai et al., 2023*; *Png et al., 2012*; *Zheng et al., 2018*; *Asgari et al., 2022*; *Macallan, 1999*), studies considering all

these components simultaneously remain scarce (*McHenry et al., 2021*; *Xu et al., 2025*; *Ogarkov et al., 2012*).

Symptomatic TB has a wide spectrum of severity, and studies conducted before anti-TB drugs became available suggest that mild disease is associated with higher odds of natural recovery (*Leavitt et al., 2024*; *Alling and Bosworth, 1960*). If mortality and natural recovery from symptomatic TB vary depending on the spectrum of disease, conceivably human populations exhibit genetic variation underlying disease severity. Furthermore, given that humans and the MTBC have coexisted for millennia, some of this variation could be linked to human ancestry. Here, we characterized the genetic variation including the genetic ancestry of a cohort of TB patients from Dar es Salaam, Tanzania, the phylogenetic lineage of their correspondent MTBC isolate, and investigated the association of both with TB disease severity while accounting for demographic, social–economical, and environmental variables.

## Results

### Genetic ancestry of Tanzanian TB patients

Genetic ancestries were estimated for 7479 individuals from 249 populations (*Figure 1—figure supplement 1*) including 1444 Tanzanian TB patients, using the software Admixture (*Alexander et al., 2009*) with 53,255 SNPs (*Figure 1—figure supplement 2*).

The optimal number of source populations to describe our dataset was 24, based on the lowest cross-validation error (*Alexander and Lange, 2011*; *Figure 1—figure supplement 3*). For this study, we named the genetic ancestries according to the geographical distribution and/or ethnicity of the reference populations that they are most prevalent in. The genetic ancestry with the highest contribution among Tanzanians with a mean of 44% (maximum 68%, minimum 0%) was also the most abundant in BS people from Southern and southeastern Africa (e.g. the Ronga population in *Figure 1*), and hence hereafter, we will refer to this ancestry as 'Southeastern BS' (*Figure 1*). The second most common genetic ancestry with a mean of 22% (maximum 42%, minimum 6%) in the Tanzanian TB patients was most common among Kenyans (e.g. the Luhya population in *Figure 1*, 1000G and HGDP, see methods), and will be referred to as the 'Eastern BS' genetic ancestry. Additionally, the Tanzanian TB patients had a mean of 9% (maximum 53%, minimum 0%) of a genetic ancestry that was most common among BS populations from western Central Africa (e.g. the Eviya population in *Figure 1*); we will refer to it as the 'Western BS' genetic ancestry. Furthermore, the Tanzanian TB patients contained on average 4% of a genetic ancestry most abundant among Nigerians represented by the Esan population ('Nigerian' genetic ancestry in *Figure 1*), and 4% of a genetic ancestry most abundant in people from Chad and Sudan (represented by the Nuba population in *Figure 1*). In addition, the genetic ancestry of the Tanzanian TB patients was composed of 3% of a genetic ancestry most prevalent in people from Western Africa (Gambia and Senegal represented by the Senegal Bedik population, in *Figure 1*), as well as 3% of a genetic ancestry most prevalent in individuals from western Africa rainforest hunter–gatherer populations (Bezan population in *Figure 1*). A mean of 2% belonged to a genetic ancestry most common among Bedouin individuals (represented by the Yemenite Jew population in *Figure 1*). The proportions of the remaining genetic ancestries were all smaller than 2% (*Figure 1—figure supplement 4*, admixture plots for all African populations included can be found in *Figure 1—figure supplement 5*). Finally, most Tanzanian TB patients had little non-African genetic ancestry (mean 5%, maximum 65%), with only 14 patients (~1%) showing more than 30% non-African genetic ancestry. In summary, the ancestry of Tanzanian TB patients was, for the most part, a mixture of three different Bantu components. Thus, for the remaining sections, we will focus on the ancestries termed Eastern BS (eBS), Southeastern BS (seBS), and Western BS (wBS).

### Insights into the BS genetic ancestries in Africa and Tanzania

Compiling several datasets, including many different BS populations, allowed for a closer look at the distribution of BS ancestries across African populations. Like Tanzanians, the populations from the neighboring Kenya and Mozambique showed strong contributions of BS ancestries resulting from different admixture events (*Figure 1*). While the eBS genetic ancestry was highest in Kenya and Tanzania, decreasing from there to the south and to the west of the continent (*Figure 2A*), the seBS genetic ancestry generally increased toward the south and decreased toward the west as observed by

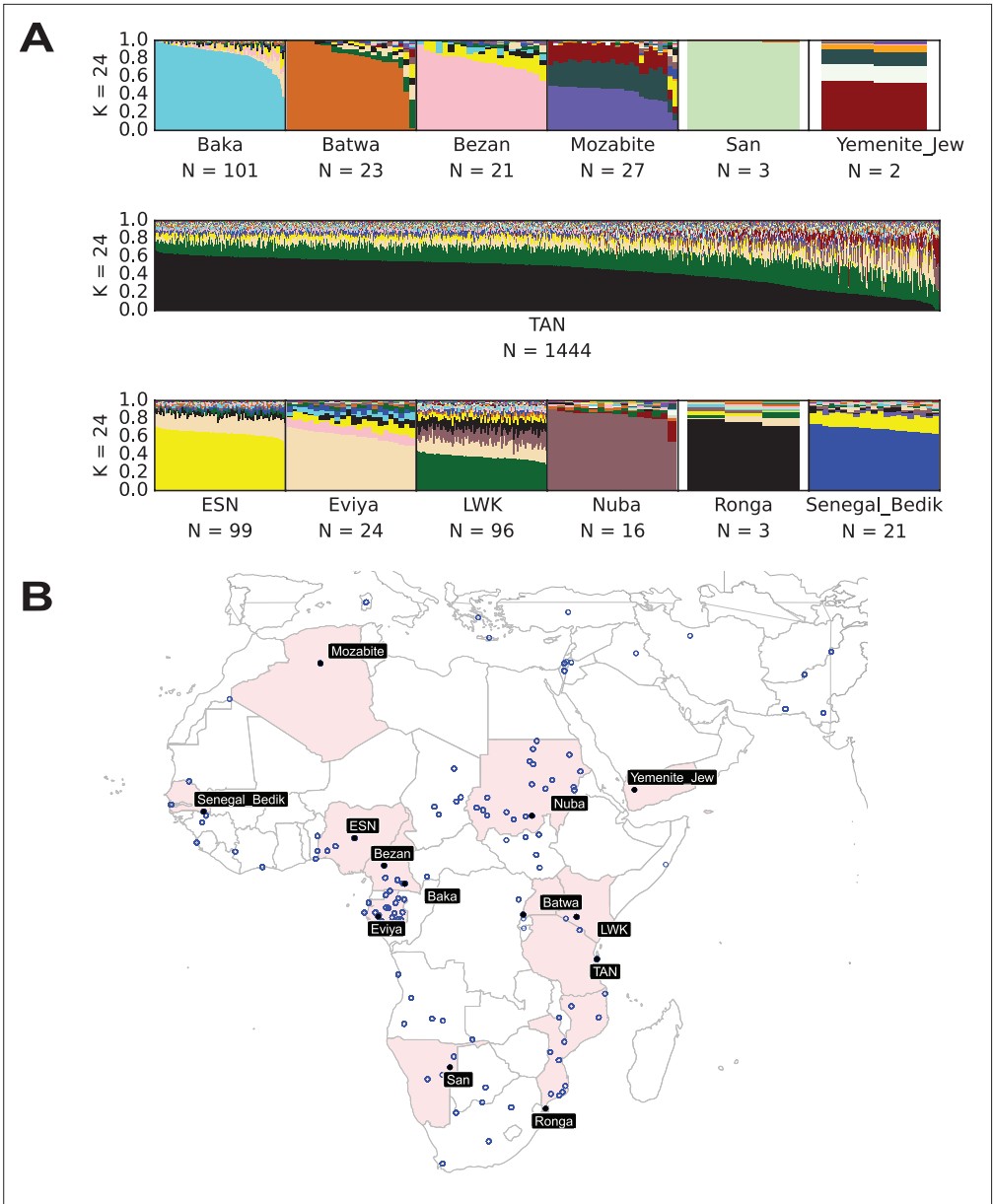

**Figure 1.** Genetic ancestry analyses of Tanzanian TB patients. (**A**) Genetic ancestry proportions of the 1444 Tanzanian TB patients and representative human populations who shared at least 1% of their most common genetic ancestry with the Tanzanians for *K* = 24 (ESN: Esan from Nigeria (1000G), LWK: Luhya from Kenya (1000G)). For all populations included in our study, see *Figure 1—figure supplement 1* for their geographic distribution and *Figure 1—figure supplement 5* for the ancestry composition of all African populations included in this study. (**B**) The geographical location of the representative populations shown in A is depicted with black circles, and the corresponding country is highlighted. The remaining African populations included in the analysis are represented by blue circles.

The online version of this article includes the following figure supplement(s) for figure 1:

**Figure supplement 1.** Populations included in the admixture analysis.

**Figure supplement 2.** The first two principal component analysis (PCA) components for all African populations included in this study (*n* = 116).

**Figure supplement 3.** Boxplots of cross-validation errors for values of *K* between 2 and 29 resulting from 15 runs.

**Figure supplement 4.** Boxplots of the proportions of the 24 genetic ancestries among the Tanzanian TB patients.

**Figure supplement 5.** Ancestry plots for all African populations included (*K* = 24).

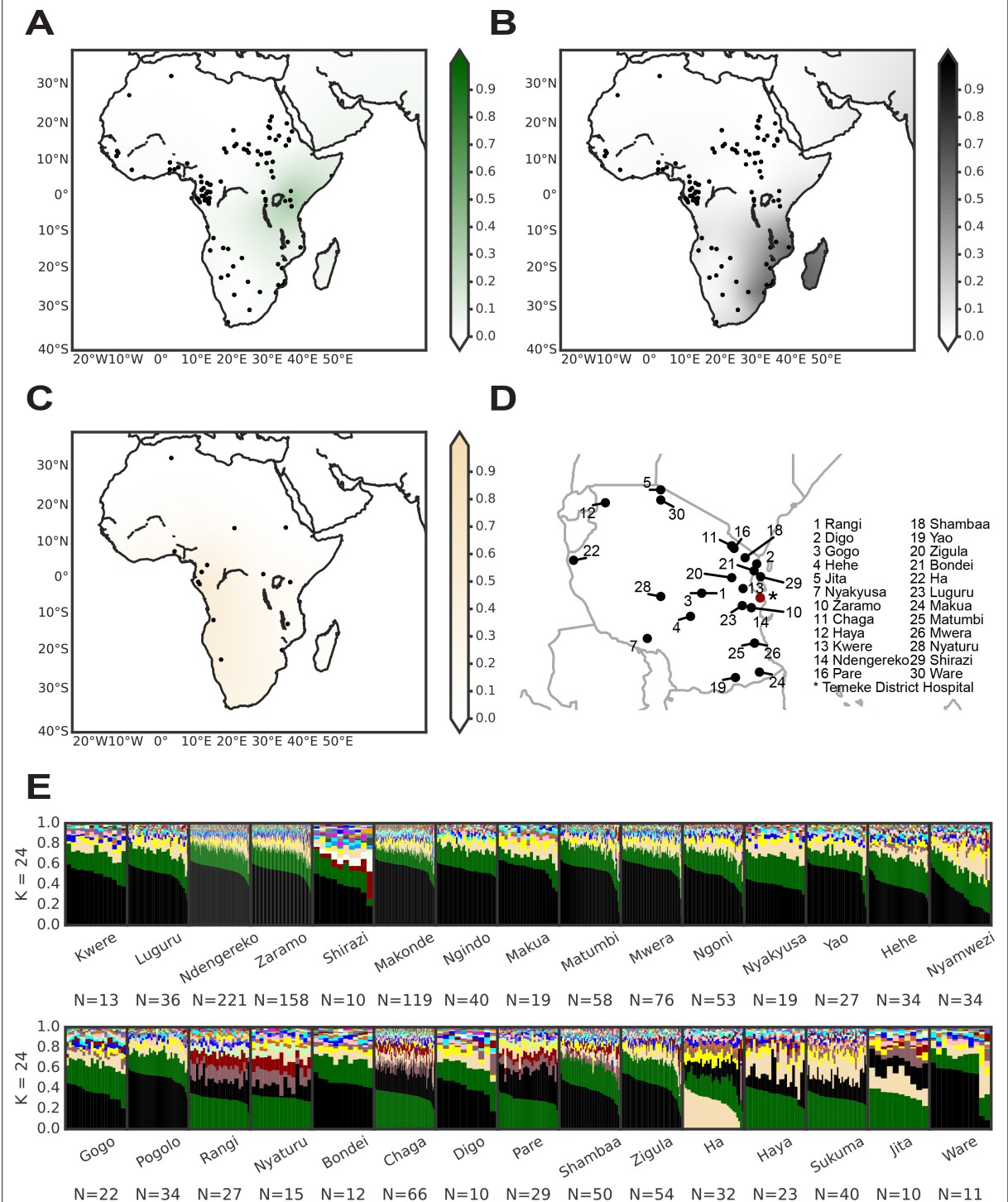

**Figure 2.** Spatial visualizations of the Bantu-speaking (BS) genetic ancestries and the genetic ancestries of the different self-identified ethnic groups among the TB patients in Tanzania. The genetic ancestry was inferred by admixture with $K = 24$, and the interpolation of the ancestries was performed by using the pykrige module in Python (see methods). (**A**) eBS genetic ancestry, (**B**) seBS genetic ancestry, and (**C**) wBS genetic ancestry. The populations included for spatial interpolations are marked with a black dot on the maps. The maps were created using the basemap module in Python. (**D**) Geographical origin of the ethnic groups among our TB patient cohort. The Temeke District hospital in Dar es Salaam where the patients were recruited is marked with a red point. Note that for some ethnic groups, no geographical origin could be identified (*Supplementary file 1*). (**E**) Ancestry plots for the different ethnic groups with at least 10 patients from our TB patient cohort.

The online version of this article includes the following figure supplement(s) for figure 2:

**Figure supplement 1.** Heatmap showing the correlations between the genetic ancestries and geographical location.

others (*Figure 2B*; *Semo et al., 2020*). The wBS genetic ancestry was mainly seen in BS populations from Gabon and Cameroon, as well as in South African populations (*Figure 2C*). At a continental-wide level, the geographical distribution and the genetic distances of the human populations analyzed were significantly correlated (Mantel test: veridical correlation = 0.18, p-value <0.001, *Figure 2— figure supplement 1*).

Even though all our TB patients were recruited in Dar es Salaam, we found them to belong to a variety of self-defined ethnic groups linked to different geographical regions within Tanzania (*Figure 2D*). Even at this smaller scale, we found a significant correlation between the geographic distance of the self-defined ethnic groups of our TB patients and their genetic distances (Mantel test: veridical correlation = 0.12, p-value <0.001). The geographical structuring of human genetic diversity within Tanzania was further supported by our finding that the seBS genetic ancestry increased from West to East, and from North to South. The eBS genetic ancestry increased from South to North and decreased from West to East. The wBS genetic ancestry increased to the North and decreased to the East (*Figure 2—figure supplement 1A*).

## The MTBC genotypes circulating in Tanzania and their association with TB disease severity

In previous work (*Zwyer et al., 2023*), we investigated the MTBC genotypes causing TB in the patient cohort analyzed here. We found a high diversity of MTBC genotypes, with approximately half of the TB cases caused by four main MTBC genotypes estimated to have been introduced into Tanzania starting 320 years ago. After adding the genomic information of an additional 389 MTBC isolates to our dataset (new total $N$ = 1471), the prevalence of the four dominant MTBC genotypes was very similar to our previous findings (*Zwyer et al., 2023*). The most frequent genotype within L3.1.1 (referred to as 'Introduction 10') contributed 39% of the current TB cases, followed by a genotype within L1.1.2 ('Introduction 9') with 9%, a genotype within L4.3.4 ('Introduction 5') with 6%, and a genotype within L2.2.1 ('Introduction 1') with 5%. The remaining TB cases were caused by a variety of other genotypes within L1–L4 but occurred at frequencies of 1% or less (*Zwyer et al., 2023*). Despite a 36% increase in sample size compared to our previous analysis (*Zwyer et al., 2023*), we still found no evidence of an association between the different MTBC genotypes circulating in Dar es Salaam and TB disease severity using as proxies; X-ray scores (mild or severe), bacterial load (inferred from GeneXpert cycle threshold), and TB-score (mild or severe) (*Table 1*, *Appendix 1—figure 1*, Table 3) (see Methods).

## Relationship between the human and MTBC population structures

We previously found that the four dominant MTBC genotypes in Dar es Salaam differed in their transmission rate and in the duration of the infectious period (*Zwyer et al., 2023*). Here, we assessed whether there might be an additional host genetic contribution to these differences. We first compared

**Table 1.** Human and bacterial genotypes by the severity measures.

| | | TB-score | | Lung damage (X-ray score) | | | | | | Bacterial load (Ct-value) | | |
| | Levels | Total N (%) | Missing N (%) | Severe N (%) | Mild N (%) | Total N | Missing N | Severe | Mild | Not available | Total N | Missing N | Mean (SD) |
|---|---|---|---|---|---|---|---|---|---|---|---|---|---|
| Total N (%) | | | | 624 (33) | 1280 (67) | | | 177 (9) | 849 (45) | 878 (46) | | | |
| MTBC genotype | Other | 1471 (77.3) | 433 | 207 (42) | 406 (41) | 764 (74.5) | 262 | 51 (39) | 269 (43) | 293 (41) | 863 (78.3) | 239 | 19.4 (4.9) |
| | L2.2.1 – Intro 1 | | | 22 (4) | 49 (5) | | | 7 (5) | 28 (4) | 36 (5) | | | 20.5 (5.1) |
| | L3.1.1 – Intro 10 | | | 184 (38) | 388 (40) | | | 59 (45) | 239 (38) | 274 (39) | | | 19.1 (4.7) |
| | L4.3.4 – Intro 5 | | | 26 (5) | 60 (6) | | | 4 (3) | 38 (6) | 44 (6) | | | 19.5 (4.1) |
| | L1.1.2 – Intro 9 | | | 51 (10) | 78 (8) | | | 11 (8) | 58 (9) | 60 (8) | | | 19.1 (4.7) |
| seBS Bantu | Mean (SD) | 1442 (75.7) | 462 | 0.44 (0.2) | 0.44 (0.2) | 840 (81.9) | 186 | 0.45 (0.1) | 0.43 (0.2) | 0.45 (0.2) | 810 (73.5) | 292 | 19.9 (5.2) |
| eBS | Mean (SD) | 1442 (75.7) | 462 | 0.23 (0.1) | 0.22 (0.1) | 840 (81.9) | 186 | 0.22 (0.1) | 0.23 (0.1) | 0.22 (0.1) | 810 (73.5) | 292 | 19.9 (5.2) |
| wBS | Mean (SD) | 1442 (75.7) | 462 | 0.08 (0.1) | 0.09 (0.1) | 840 (81.9) | 186 | 0.08 (0.1) | 0.09 (0.1) | 0.08 (0.1) | 810 (73.5) | 292 | 19.9 (5.2) |
| Other ancestry | Mean (SD) | 1442 (75.7) | 462 | 0.25 (0.1) | 0.25 (0.1) | 840 (81.9) | 186 | 0.25 (0.1) | 0.25 (0.1) | 0.25 (0.1) | 810 (73.5) | 292 | 19.9 (5.2) |

**Table 2.** Characteristics of MTBC genotypes for all patients with either human or bacterial genetic data available.

| | N (%)* | Missing N | Levels | Other genotype | L2.2.1 – Intro 1 | L3.1.1 – Intro 10 | L4.3.4 – Intro 5 | L1.1.2 – Intro 9 | No bacterial data available |
|---|---|---|---|---|---|---|---|---|---|
| Total N (%) | | | | 613 (32) | 71 (4) | 572 (30) | 86 (5) | 129 (7) | 433 (23) |
| Sex[†] | 1471 (100.0) | 0 | Male (%) | 424 (69) | 52 (73) | 425 (74) | 55 (64) | 87 (67) | 302 (70) |
| | | | Female (%) | 189 (31) | 19 (27) | 147 (26) | 31 (36) | 42 (33) | 131 (30) |
| Age in years | 1471 (100.0) | 0 | Median (IQR) | 33.0 (28.0–40.0) | 31.0 (24.5–38.0) | 33.0 (26.0–41.0) | 31.5 (27.0–38.8) | 35.0 (26.0–45.0) | 35.0 (27.0–43.0) |
| HIV status[†] | 1452 (98.7) | 19 | Infected (%) | 90 (15) | 10 (14) | 103 (18) | 19 (22) | 31 (25) | 97 (23) |
| Smoker[†] | 1443 (98.1) | 28 | Yes (%) | 127 (21) | 18 (26) | 149 (27) | 11 (13) | 26 (20) | 97 (23) |
| Cough duration (weeks) | 1454 (98.8) | 17 | Median (IQR) | | 4.0 (3.0–4.0) | 4.0 (3.0–4.0) | 3.5 (2.2–4.0) | 4.0 (2.0–5.0) | 3.0 (2.0–4.0) |
| Socioeconomic status | 1452 (98.7) | 19 | Median (IQR) | 80.0 (50.0–125.0) | 75.0 (50.0–125.0) | 87.5 (50.0–126.2) | 85.4 (50.0–122.9) | 75.0 (50.0–125.0) | 83.3 (50.0–133.3) |
| Education[†] | 1471 (100.0) | 0 | No education (%) | 90 (14.7) | 5 (7.0) | 76 (13.3) | 10 (11.6) | 15 (11.6) | 76 (17.6) |
| | | | Primary (%) | 391 (63.8) | 54 (76.1) | 390 (68.2) | 61 (70.9) | 92 (71.3) | 273 (63.0) |
| | | | Secondary (%) | 105 (17.1) | 12 (16.9) | 90 (15.7) | 13 (15.1) | 18 (14.0) | 64 (14.8) |
| | | | University (%) | 27 (4.4) | 0 (0.0) | 16 (2.8) | 2 (2.3) | 4 (3.1) | 20 (4.6) |
| Malnutrition[†] | 1471 (100.0) | 0 | Yes (%) | 83 (13.5) | 9 (12.7) | 71 (12.4) | 11 (12.8) | 13 (10.1) | 56 (12.9) |
| Relapse/reinfection[†] | 1447 (98.4) | 24 | Yes (%) | 9 (1.5) | 4 (5.6) | 18 (3.2) | 2 (2.4) | 3 (2.3) | 12 (2.8) |
| Drug resistance status[†] | 1471 (100.0) | 0 | Susceptible (%) | 574 (93.6) | 70 (98.6) | 556 (97.2) | 80 (93.0) | 119 (92.2) | 0 |
| | | | INH-Mono (%) | 37 (6.0) | 1 (1.4) | 16 (2.8) | 6 (7.0) | 10 (7.8) | 0 |
| | | | MDR (%) | 0 (0.0) | 0 (0.0) | 0 (0.0) | 0 (0.0) | 2 (0.3) | 0 |
| seBS | 1009 (68.6) | 462 | Median (IQR) | 0.5 (0.3–0.6) | 0.5 (0.4–0.5) | 0.5 (0.4–0.6) | 0.5 (0.4–0.6) | 0.5 (0.4–0.6) | 0.5 (0.3–0.6) |
| eBS | 1009 (68.6) | 462 | Median (IQR) | 0.2 (0.2–0.3) | 0.2 (0.2–0.3) | 0.2 (0.2–0.3) | 0.2 (0.2–0.3) | 0.2 (0.2–0.3) | 0.2 (0.2–0.3) |
| wBS | 1009 (68.6) | 462 | Median (IQR) | 0.1 (0.0–0.1) | 0.1 (0.1–0.1) | 0.1 (0.0–0.1) | 0.1 (0.0–0.1) | 0.1 (0.0–0.1) | 0.1 (0.0–0.1) |
| Other ancestry | 1009 (68.6) | 462 | Median (IQR) | 0.2 (0.2–0.3) | 0.2 (0.2–0.3) | 0.2 (0.2–0.3) | 0.2 (0.2–0.2) | 0.2 (0.2–0.2) | 0.2 (0.2–0.3) |

*The column N (%) indicates the total number of patients with bacterial genetic data available that contained a value for the respective variable.

[†]The percentage of an MTBC genotype that has the respective characteristic (e.g. percentage of males among patients infected with an Intro 1 MTBC genotype).

the genetic ancestry proportions between patients infected with the four dominant genotypes and then tested whether patients who were genetically more closely related were infected with MTBC genotypes that were also more closely related as would be expected from a co-evolutionary process (**Brites and Gagneux, 2015**). However, the human genetic ancestry proportions differed only marginally between the TB patients infected by the four main MTBC genotypes (**Table 2**). Moreover, there was no correlation between the human and bacterial genetic distances (Mantel test: veridical correlation = –0.02, p-value = 0.85). Taken together, we found no statistically significant relationship between the human and bacterial genetic population structure in Dar es Salaam. These results also suggest

that the genetic composition of this human population is unlikely to have a measurable effect on the differences in bacterial transmission rate and duration of the infectious period reported previously (*Zwyer et al., 2023*).

## Association of human genetic ancestry with TB disease severity

In a previous publication on the same cohort, we found that the bacterial genetic background could not explain the differences observed in TB disease severity (*Zwyer et al., 2023*). Since TB disease is shaped by the bacterial and the human genetic background as well as environmental factors (*Comas and Gagneux, 2009*), we next investigated whether human genetic ancestry could have contributed to the differences in disease severity observed between our TB patients. We assessed the associations between ancestry and the three proxies of TB severity in HIV-negative patients using logistic regression models. We included the three human genetic ancestries with the highest proportions among the Tanzanian TB patients (seBS, eBS, and wBS) as covariates, together with age, sex, smoking, socioeconomic status, TB category (relapse or not), malnutrition, education level, and drug resistance status to control for potential confounding. We found no evidence of an association between human genetic ancestry and any of these three proxies of TB disease severity (*Table 1*). We repeated the analysis by adding cough duration as a covariate to possibly account for the disease duration before recruitment, but the results remained unchanged. Additionally, we conducted a GWAS to test for associations

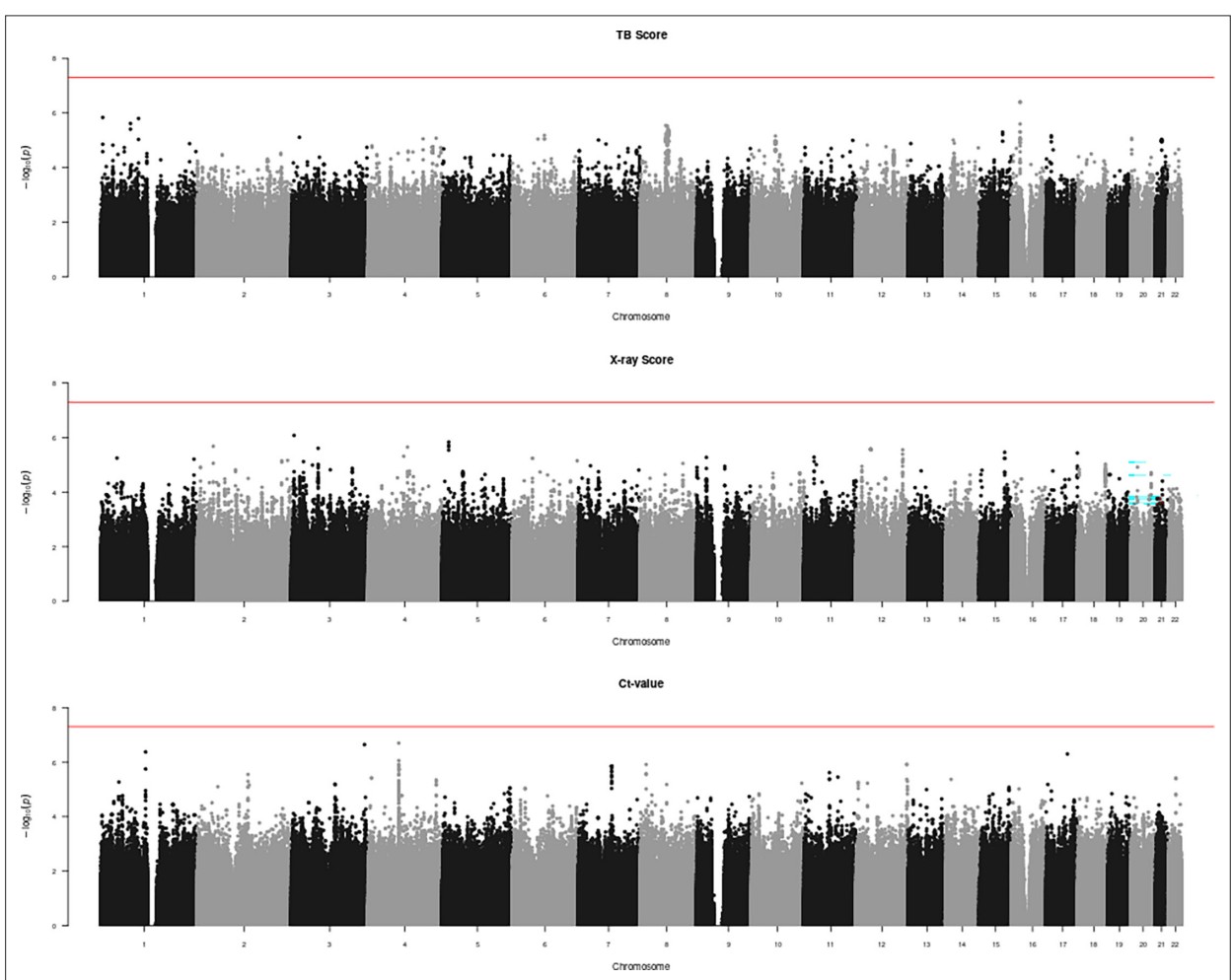

**Figure 3.** Manhattan plot for genome-wide association study (GWAS) conducted using (**A–C**) TB-score, X-ray score, and Ct-value. Red line indicates GWAS significance threshold of 5e−8.

The online version of this article includes the following figure supplement(s) for figure 3:

**Figure supplement 1.** QQ plot and genomic inflation factor for genome-wide association study (GWAS) conducted using (**A–C**) TB-score, X-ray score, and Ct-value.

between specific human genetic variants and the three proxies of TB disease severity. We included the top three human genetic principal components (PCs), HIV status, and the same covariates as for the regression described above. No evidence of an association was found (*Figure 3*).

## The combined effect of human and MTBC genetic diversity on TB disease severity

For a subset of 1000 TB patients, we had both an MTBC genome and a human genome or genotype available. Genetic interactions between the host and the pathogen have been shown to affect TB severity in other settings (*Asgari et al., 2022*; *McHenry et al., 2020a*). To test for potential interactions

**Table 3.** Estimated associations between disease severity, human genetic ancestry, and bacterial genotype. Three variables as proxies for disease severity were included: lung damage (mild versus severe), TB-score (mild versus severe), bacterial load (continuous, $\log_{10}$ transformed). Binomial logistic regressions were performed on the data of HIV-negative patients and adjusting was done for age, sex, smoking, socioeconomic status, level of education, malnutrition, TB type (relapse or new infection), and drug resistance status by including these variables in the model. For the ancestries and the interactions, the p-values were retrieved by performing a likelihood ratio test comparing a model including the ancestries and interactions to a model without them. This table combines the results of two logistic regressions per disease severity measure, one including an interaction and one without. The ancestries were transformed and categorized (see Methods) with category 1 comprising the lowest amount of the respective ancestry and category 3 (4 in the case of wBS) the highest amount.

| | | Disease severity measure | | | | | |
|---|---|---|---|---|---|---|---|
| | | Lung damage | | TB-score | | Bacterial load (Ct-value) | |
| | | OR adjusted | p-value adjusted | OR adjusted | p-value adjusted | OR adjusted | p-value adjusted |
| MTBC genotype* | L3.1.1 – Introduction 10 | 1.60 (0.95–2.67) | 0.07 | 1.00 (0.72–1.40) | 0.98 | 0.99 (0.97–1.00) | 0.13 |
| | Other genotypes | 1 | | 1 | | 1.00 | |
| Human ancestry† | seBS category 3 | 2.29 (1.04–5.42) | 0.19 | 1.06 (0.66–1.69) | 0.17 | 1.00 (0.98–1.03) | 0.13 |
| | seBS category 2 | 2.40 (1.15–5.41) | | 0.62 (0.40–0.94) | | 1.02 (0.99–1.04) | |
| | seBS category 1 | 1 | | 1 | | 1.00 | |
| | eBS category 3 | 1.30 (0.51–3.79) | | 1.09 (0.64–1.84) | | 1.02 (0.99–1.05) | |
| | eBS category 2 | 1.50 (0.61–4.28) | | 1.17 (0.69–1.94) | | 1.00 (0.97–1.03) | |
| | eBS category 1 | 1 | | 1 | | 1.00 | |
| | wBS category 4 | 0.58 (0.23–1.53) | | 0.94 (0.49–1.76) | | 1.02 (0.98–1.06) | |
| | wBS category 3 | 0.54 (0.21–1.46) | | 1.09 (0.57–2.04) | | 1.03 (0.99–1.07) | |
| | wBS category 2 | 0.73 (0.26–2.15) | | 1.03 (0.50–2.09) | | 1.03 (0.99–1.08) | |
| | wBS category 1 | 1 | | 1 | | 1.00 | |
| Interaction† | seBS category 3* L3.1.1 – Intro 10 | 1.05 (0.20–5.43) | 0.06[1] | 1.05 (0.40–2.71) | 0.92 | 0.99 (0.94–1.04) | 0.83 |
| | seBS category 2* L3.1.1 – Intro 10 | 0.39 (0.08–1.94) | | 1.29 (0.52–3.15) | | 0.98 (0.93–1.03) | |
| | seBS category 1* L3.1.1 – Intro 10 | 1 | | 1 | | 1.00 | |
| | eBS category 3* L3.1.1 – Intro 10 | 8.32 (0.91–193.14) | | 1.45 (0.48–4.34) | | 1.01 (0.94–1.07) | |
| | eBS category 2* L3.1.1 – Intro 10 | 6.38 (0.71–146.86) | | 1.25 (0.42–3.70) | | 1.02 (0.96–1.09) | |
| | eBS category 1* L3.1.1 – Intro 10 | 1 | | 1 | | 1.00 | |
| | wBS category 4* L3.1.1 – Intro 10 | 0.17 (0.02–1.23) | | 1.79 (0.49–6.56) | | 0.97 (0.90–1.05) | |
| | wBS category 3* L3.1.1 – Intro 10 | 0.33 (0.04–2.51) | | 1.48 (0.40–5.51) | | 0.99 (0.92–1.07) | |
| | wBS category 2* L3.1.1 – Intro 10 | 0.87 (0.09–8.06) | | 2.40 (0.53–11.05) | | 0.98 (0.90–1.07) | |
| | wBS category 1* L3.1.1 – Intro 10 | 1 | | 1 | | 1.00 | |

* The odds ratio for genotype represents the odds of severe disease for Introduction 10 compared to the odds for other genotypes.

†The odds ratios represent the estimated multiple in the odds of severe disease for a one-unit increase in the additive log-transformed ancestry variable.

between human and bacterial diversity on TB severity, we added to the ancestry logistic regression models described in the previous section the most common MTBC genotype as an additional explanatory variable (L3.1.1 – Introduction 10), as well as the interaction between human ancestry and MTBC genotype. We only tested L3.1.1 – Introduction 10, since the numbers of patients with the other genotypes were too few for meaningful testing. However, we found no evidence for any interaction between the MTBC genotype and human ancestry influencing TB disease severity in this patient population (*Table 3*).

## Discussion

In this study, we analyzed the genetic ancestry of TB patients, the MTBC diversity underlying their TB infection, and estimated the associations of both with disease severity in Dar es Salaam, Tanzania. We found a strong component of BS genetic ancestries among the Tanzanian TB patients, similar to those of neighboring populations from Mozambique and Kenya, and little non-African genetic ancestry. Genetic ancestry proportions did not differ between patients infected with different MTBC genotypes. There was no evidence that the patient genetic ancestry or the MTBC genotype on their own, nor their interaction, had any effect on TB disease severity.

Despite the fact that Tanzania is one of the few countries in sub-Saharan Africa where all four main African linguistic groups co-exist (*Tishkoff et al., 2009*), and that its largest city and economic capital, Dar es Salaam, is strongly influenced by different human populations from within and outside Africa, our cohort of TB patients mostly comprised BS ethnicities. Comparing this TB patient cohort to a large number of modern human populations revealed major components of eBS and seBS genetic ancestries. This genetic population structure probably resulted from several admixture events estimated to have happened between 1500 and 150 years ago, between local populations and BS populations who migrated from West Africa to the East and South of the continent (*Patin et al., 2017*). The TB patients investigated here were recruited in one district hospital of Dar es Salaam. Yet, we found the genetic distances between the patients to be correlated with the original geographic range of their self-identified ethnicities, suggesting that the corresponding human populations are not fully admixed. The population of Dar es Salaam has increased by several millions in the last 40 years, mainly as a result of immigration from rural parts of Tanzania (*Llc, 2026*). Thus, our findings suggest that our TB patient population mostly represents recent migrants to Dar es Salaam from other regions of Tanzania. Moreover, we found little evidence of Eurasian genomic influence in the TB patient population (on average 5% genetic ancestry). This is in strong contrast to coastal Swahili populations, as recent findings comparing modern and medieval Swahili people revealed large components of genetic ancestry derived from exchanges between local East African BS populations and people from India, Persia, and Arabia, starting as early as 1000 AD (*Brielle et al., 2023*). We conclude that our TB patient population does not represent the full spectrum of human genetic diversity in Tanzania.

In contrast to the genetic ancestry of the TB patients, we found that the MTBC genotypes infecting these patients descend from multiple historical introductions, which mainly resulted from the human exchanges that took place across the Indian Ocean during the last few centuries (*Zwyer et al., 2023*). Some of these recently introduced MTBC genotypes became dominant, in particular the MTBC genotype L3.1.1 – Introduction 10, which caused TB in approximately 40% of our patients. These strains descended from an introduction that occurred approximately 300 years ago from South or Central Asia to East Africa (*Zwyer et al., 2023*). Hence, while our TB patient population reflects little historical gene flow from non-African populations, the underlying MTBC diversity indicates that the MTBC genotypes introduced from outside successfully spread in this newly encountered host population, eventually outcompeting native MTBC genotypes (*Comas et al., 2015*).

We previously reported that in this TB patient cohort, some of the dominant MTBC genotypes had a higher transmission rate than others, while some other MTBC genotypes induced patients to remain infectious for longer (*Zwyer et al., 2023*). Based on the similar proportions of MTBC genotypes among self-reported ethnic groups we observed at the time, we had already hypothesized that human genetic heterogeneity of the host population is unlikely to be responsible for those differences (*Zwyer et al., 2023*). Here we formally addressed this hypothesis and found that there was no evidence that the human genetic ancestry proportions differed between patients infected with different MTBC genotypes in our cohort. This finding is consistent with the notion that the differences in epidemiological parameters we reported previously are mainly determined by the pathogen genotype.

Disease severity is one aspect of the clinical presentation of TB with a direct impact on patient mortality and morbidity, as well as on pathogen transmission, as it influences patient infectiousness. It is thus likely that both host and pathogen genetic characteristics can modulate TB disease severity (*Brites and Gagneux, 2015*). Experimental infections in various animal models suggest that different MTBC strains vary in virulence (*López et al., 2003*; *Aguilar L et al., 2010*; *Dormans et al., 2004*). However, *in clinico* evidence for differences in disease severity caused by different MTBC genotypes is inconsistent (*Coscolla and Gagneux, 2010*; *Stanley et al., 2024*). We found no evidence of differences in disease severity at the point of care caused by the different MTBC genotypes in our study. Moreover, we did not observe any association between human genetic ancestry and disease severity, which is in contrast to a recent study from Peru, where human genetic ancestry was found to influence progression to active TB (*Asgari et al., 2022*).

Our previous work found evidence of such genetic interactions when considering the complete genomes of the TB patients and their infecting MTBC strain (*Xu et al., 2025*) and identified associations between human and pathogen variants. Such associations reflect host–pathogen genetic interactions that determine susceptibility to symptomatic TB or intra-host selection during mycobacterial replication. Here, in the context of co-evolutionary history between humans and MTBC, we specifically tested whether an interaction between the main human ancestry components and being infected with the most dominant MTBC strain in Dar es Salaam could explain the variability in TB disease severity. However, we did not find evidence of such an effect. Others have reported an association between TB disease severity, a particular bacterial genotype, and a particular human SNP in Uganda but did not explicitly link this human SNP to a particular human genetic ancestry (*McHenry et al., 2020a*; *McHenry et al., 2020b*). Several factors could account for the lack of effect we observed. First, our patient population was relatively genetically homogeneous, given that the different BS components represent populations with only moderate levels of genetic differentiation (*Cavalli-Sforza et al., 1994*). Second, there is likely to be selection bias in our cohort since only patients presenting at the clinic were recruited. The disease severity measures included in this study mainly reflect disease stages at which patients felt ill enough to go to the hospital, that is we did not consider intermediate, more contrasting disease states that are known to occur between infection and the development of symptoms (*Frascella et al., 2021*). To at least partially account for that, we included the number of weeks a patient was coughing as a covariate in our analyses. Third, the lack of a measurable interaction between host genetic ancestry and MTBC genotype could reflect the relatively recent presence of these MTBC genotypes in Tanzania, and the distinct (i.e. allopatric) geographical origins of the host and pathogen populations. This indicates that none of the ancestral human populations that compose modern Tanzanians has lived in sympatry with the ancestors of the modern MTBC genotypes that circulate in Dar es Salaam today. With the exception of West Africa, where the geographically restricted West-African MTBC lineages L5 and L6 remain an important cause of human TB (*Silva et al., 2022*), the situation in Tanzania might be representative of the TB epidemics in many African countries, as evidence suggests that many MTBC genotypes dominating the continent today have been introduced from outside Africa recently in the history of human populations (*Comas et al., 2015*; *Rutaihwa et al., 2019*; *Menardo et al., 2021*; *Chihota et al., 2018*).

In conclusion, our study shows that the TB patients from Dar es Salaam were mainly of BS genetic ancestry reflecting limited Eurasian genetic influx. Neither the human genetic ancestry nor the MTBC genotype alone, nor their interaction, was associated with TB disease severity. Our results highlight the dominant role of social and environmental factors in human TB in Tanzania.

## Methods

### Study population

This study is based on a previously described dataset (*Xu et al., 2025*; *Zwyer et al., 2023*). Briefly, adult active TB patients with pulmonary disease (sputum smear-positive and GeneXpert-positive) were recruited between November 2013 and June 2022 at the Temeke District Hospital in Dar es Salaam, Tanzania, when they first presented for care. Sputum and blood samples were collected from each patient to extract DNA for sequencing of the MTBC strain and genotyping or whole-genome sequencing (WGS) of the patient. Additionally, clinical and sociodemographic information was obtained from every patient. In total, there was either human or bacterial data available for

1904 patients. Of those, 1444 patients had human genetic data and 1471 had bacterial genetic data available, respectively. A total of 1000 patients had both types of data available after quality-based filtering. The geographical locations of the self-indicated ethnic group of each patient were retrieved by searching for the original region of the respective ethnic group, and if they originated from a single region, the geographic coordinates according to Wikipedia were taken. If two neighboring regions were among the origins, then a random location between the two regions was taken as surrogate (*Supplementary file 1*).

## Bacterial and human sample processing

The MTBC bacteria were cultured on solid Löwenstein-Jensen media at the TB laboratory of the Ifakara Health Institute in Bagamoyo, Tanzania. Before 2018, the MTBC isolates were transferred to Switzerland for DNA extraction and later, DNA extraction was carried out in Bagamoyo. Bacterial WGS was done using the Illumina short-read technology at the Department of Biosystems Science and Engineering of ETH Zurich in Basel (DBSSE). Human WGS was done at the Health 2030 Genome Center in Geneva, Switzerland, using an Illumina NovaSeq 6000 sequencer. The human genotyping was done at the iGE3 Genomics platform at the University of Geneva in Switzerland using the Illumina Infinium H3Africa genotyping microarray (Version 2; https://chipinfo.h3abionet.org) plus custom Tanzanian-specific SNP add-ons (*Xu et al., 2022*).

## Human genetic data

The processing of the human genetic data has been described in detail by *Xu et al., 2025*. Briefly, we used the GRCh38 as a human reference genome to map the WGS reads of 118 patients using BWA aligner (v0.7.17) (*Li and Durbin, 2009*). Duplicate reads were then marked with the markduplicates module of Picard (v2.8.14) (*broadinstitute, 2019*). Variant calling was first done for each sample individually following GATK best practices for germline short variant discovery. Samples with a coverage below 5 were excluded, followed by a joint calling of the variants. A Quality Score Recalibration (VQSR) based filter was applied (real sensitivity of 99.7, excess heterozygosity of 54.69) and samples with more than 50% missing genotype calls were removed.

For the genotyping data, we used the Illumina GenomeStudio software (v2.0.5, https://support.illumina.com/array/array_software/genomestudio/downloads.html) to analyze the raw microarray data. Samples with a low quality, that were badly clustered, or that had a call rate below 0.97 were excluded. The PLINK format was then converted to VCF format using PLINK (v1.9) (*Chang et al., 2015*). The first round of imputation was performed with the African Genome Resources (AFGR, https://www.apcdr.org/) reference panel on the sanger imputation server with EAGLE (*Loh et al., 2016*) for phasing and positional Burrows–Wheeler transform (PBWT) (*Durbin, 2014*) for imputing. The second round was performed with a reference panel created in-house that was based on 118 patients with available whole-genome sequences (*Xu et al., 2022*) with SHAPEIT4 for phasing and Minimac3 for imputing. For each SNP, the reference panel with the highest imputation quality score was used to determine the final genotype call. SNPs with an INFO score below 0.8 were discarded.

Bcftools (v1.15) was used to merge the WGS and genotyping samples after identifying SNPs shared between the two methods that were missing in fewer than 10 samples.

## Whole-genome sequence analysis of the MTBC bacteria

We analyzed all FASTQ files using the WGS analysis pipeline described previously (*Menardo et al., 2018*). In summary, Trimmomatic (*Bolger et al., 2014*) v. 0.33 (SLIDINGWINDOW:5:20) was used to remove the Illumina adaptors and to trim low-quality reads. Only reads with a length of at least 20 bp were kept for further analysis. Overlapping paired-end reads were merged using SeqPrep v. 1.2 (*John, 2011*) (overlap size = 15). We then mapped the resulting reads to a reconstructed ancestral sequence of the MTBC (*Comas et al., 2010*) with BWA v. 0.7.13 (mem algorithm) (*Li and Durbin, 2009*). Picard v. 2.9.1 (*broadinstitute, 2025*) was then applied to mark and exclude duplicated reads. Furthermore, the RealignerTargetCreator and IndelRealigner modules of GATK v. 3.4.0 (*McKenna et al., 2010*) were used to perform local realignment of reads around INDELs. Reads having an alignment score lower than $(0.93 \times read\ length) - (read\ length \times 4 \times 0.07)$, corresponding to more than 7 mismatches per 100 bp, were excluded using Pysam v. 0.9.0 (*pysam-developers, 2026*). SNP calling was then performed with SAMtools v. 1.2 mpileup (*Li, 2011*) and VarScan v. 2.4.1 (*Koboldt et al., 2012*) with

the following thresholds: a minimum mapping quality of 20, a minimum base quality at a position of 20, minimum read depth at a position of 7 and no strand bias. Positions in repetitive regions such as PE, PPE, and PGRS genes or phages were excluded, as described in *Stucki et al., 2016*. A whole-genome Fasta file was created from the resulting VCF file. We applied some additional filters; genomes were excluded from downstream analysis if they had a sequencing coverage of lower than 15 or if they contained SNPs suggestive of different MTBC lineages (i.e. mixed infections). We identified lineages and sublineages using the SNP-based classification by *Steiner et al., 2014* and *Coll et al., 2014*, respectively. In addition, we identified drug resistance mutations published in the WHO catalogue (*World Health Organization, 2021*) and determined the respective drug resistance profile. The majority of strains were susceptible (95%), while 70 (5%) were isoniazid mono-resistant and two contained isoniazid and rifampicin resistance and were considered as multi-drug resistant.

## Identification of bacterial SNPs diagnostic for the successful MTBC introductions

We previously identified several successful MTBC introductions into Dar es Salaam (*McHenry et al., 2021*). For these, we aimed to obtain a set of diagnostic SNPs that would allow assigning MTBC strains not included in our previous study to these genotypes. For that, we merged the VCF files from the 1,082 MTBC genomes included in that previous dataset by using BCFtools (v1.9). We then used VCFtools (v0.1.16) to remove Indels and positions that were variable in less than 12 genomes (12 was the minimal threshold selected when identifying the successful introductions in our previous publication *Zwyer et al., 2023*). By using the R package VariantAnnotation (*Obenchain et al., 2014*) and a customized Python script, SNPs specific to one of the most successful introductions were extracted. To ensure the SNPs identified as markers for the introductions were specific, we also identified phylogenetic SNPs on a bigger and global dataset representing the human-adapted MTBC diversity (*Menardo et al., 2018*) and tested whether any of the phylogenetic SNPs identified for any of the successful introductions was present in any of the MTBC lineages or sublineages. We compiled a subset of 25 SNPs (*Supplementary file 2*) that we used as phylogenetic markers for the different MTBC introductions and identified strains belonging to one of the four most successful MTBC introductions (Introduction 1, Introduction 5, Introduction 9, and Introduction 10) in the expanded MTBC dataset based on these SNPs.

## Measures of TB disease severity

We used three different proxies for TB disease severity. The first one was the TB-score, which is a clinical score adapted from *Wejse et al., 2008* that consists of several signs and symptoms including the presence of fever and the body mass index (BMI). A point was given for each of the following symptoms or clinical measures if present: BMI below 18, BMI below 16, mid-upper arm circumference (MUAC) below 220, MUAC below 200, body temperature higher than 37°C, cough, hemoptysis, dyspnea, chest pain, night sweat, abnormal auscultation, and anemia. A maximum of 12 points could be achieved, and a TB-score below 6 was considered as mild and everything above as severe. As a second proxy, we assessed the amount of lung damage. Two independent radiologists assessed chest X-ray pictures of the patients and gave a Ralph score (*Ralph et al., 2010*). X-ray scores above 71 were considered severe, and everything below was considered mild. The Ralph score has been validated to grade chest X-ray severity in adult pulmonary TB patients, and 71 was the optimal cutoff point for predicting unfavorable outcome (*Li, 2011*). Furthermore, Ralph scores higher than 71 have been associated with a longer duration of symptoms, a lower BMI, and higher clinical scores (*Chakraborthy et al., 2018*). As a third proxy, we used the bacterial load in the sputum represented by the difference between the first (early cycle) and the last (late cycle) threshold during quantitative PCR (Ct-value) as determined by GeneXpert MTB/RIF assays. For each sputum sample, we took five probes, ran a quantitative PCR each, and reported the lowest Ct-value.

## Genetic ancestry analysis of TB patients

To investigate the genetic ancestry of the TB patients, we combined our dataset with the data from ten other projects: The Gambian Genome Variation Project (GGVP) (*Band et al., 2019*), the 1000 Genomes Project (1000G) (*Altshuler et al., 2015*), the Human Genome Diversity Project (HGDP) (*Bergström et al., 2020*), Simons Genome Diversity Project (*Mallick et al., 2016*), as well as data

generated by *Patin et al., 2017*; *Patin et al., 2014*, *Hollfelder et al., 2017*, *Semo et al., 2020*, *Schlebusch et al., 2012*, and *Fortes-Lima et al., 2022*. We used the GRCh37 version of all datasets. The dataset of HGDP was in GRCh38 version, and we thus did a lift over to GRCh37 using the picard (v2.26.10) (*broadinstitute, 2019*) tool LiftoverVcf. For all the datasets including only populations from one single continent, we excluded variants with a missingness of more than 10% and only included variants that did not deviate from Hardy–Weinberg equilibrium (p < 1e−5) using PLINK (version 1.9b, https://www.cog-genomics.org/plink/1.9/) (*Chang et al., 2015*). For the 1000G, SGDP, and HGDP data, we first identified variants that deviated from the Hardy–Weinberg equilibrium (p < 1e−5) in each superpopulation using PLINK (version 2.0a, https://www.cog-genomics.org/plink/2.0/) and removed them from the whole dataset. We additionally removed variants with a high missingness (>10%) from the full datasets using PLINK (version 1.9b). After extraction of 103,262 nucleotide positions common to all datasets, we merged the datasets using PLINK (version 1.9b). From the merged dataset, we removed second-degree relatives using PLINK (version 2.0a, king cutoff of 0.088) (*N* = 369) and patients from our cohort where the sex according to the genetic data did not correspond with the sex indicated in the clinical data, patients who were genetic outliers based on principal component analysis (PCA) or who did not cluster with any other African samples (*N* = 83). In addition, we removed regions of high linkage disequilibrium (https://genome.sph.umich.edu/wiki/Regions_of_high_linkage_disequilibrium_(LD)) and applied additional filters to the merged dataset (missingness >10%, minimum allele frequency of 5%, removal of sex chromosomes, variant pruning with –indep-pairwise 50 10 0.1, only biallelic positions) ending up with 53,255 positions and 7479 individuals from 249 populations.

To infer the ancestry proportions of the Tanzanian TB patients, we used ADMIXTURE (version 1.3.0) (*Alexander et al., 2009*). We estimated the number of ancestral populations (*K*) by running ADMIXTURE 15 times for each value for *K* from *K* = 2 until *K* = 29 with the option `--cv`. The `--cv` option performs fivefold cross-validation and allows identifying the value for *K* resulting in the lowest cross-validation error (*Figure 1—figure supplement 3*). The cross-validation error was lowest for *K* = 24. From the 15 runs performed with *K* = 24, we selected a representative of an output that was supported by most of the 15 runs (6/15) to extract the ancestry proportions of each individual. A PCA was performed using PLINK (version 1.9b) on all African populations included.

## Spatial interpolation of human genetic ancestry proportions

To visualize the distributions of the different patient ancestries, we performed spatial interpolation using the OrdinaryKriging function from the pykrige module in Python (variogram_model = 'linear', grid space of 500). For the eBS and seBS ancestries, we included all African populations for the interpolation. For the wBS, the interpolation failed when using all African populations with an insufficient slope, suggesting little spatial variability. Since many populations were sampled in the region with the highest proportions of wBS genetic ancestry and among them many hunter–gatherer populations containing little to no wBS genetic ancestry, we repeated the interpolation with only a subset of the non-hunter–gatherer populations.

## Correlation between distance matrices

To assess whether MTBC genotypes that are more closely related tend to infect people that are also more closely related genetically, which would be compatible with co-evolution, we investigated the correlation between the human and bacterial distance matrices. To calculate the pairwise bacterial genetic distances, alignments of variable positions where data was missing in less than 10% of the genomes were generated and used to create SNP distance matrices according to the Hamming distance (https://git.scicore.unibas.ch/TBRU/tacos). Insertions and deletions were considered as missing data. To get the human pairwise genetic distances, we calculated the Euclidean distance based on the first two PCs. When only looking at the Tanzanian TB patients, we calculated the PCs for the Tanzanians only, while for the continental dataset, we included all available African populations.

To investigate whether human populations that are geographically more distant are also genetically more distant, we calculated the correlation between the geographical and the genetic distance on an African level as well as on the level of Tanzania. For the geographical distance matrix, we calculated the Euclidean distance based on the latitude and the longitude. At the level of Tanzania, the broad geographic location of the original area of the ethnic group was considered (*Supplementary*

*file 1*), and for the continental level, the coordinates of the hospital in Temeke were taken for the Tanzanian TB patients, considering that for the other studies only the sampling locations were known. The human and bacterial genetic distance matrices as well as the human genetic and geographic distance matrices were tested for correlations using the mantel.test() function from the mantel module in Python (options: perms = 10,000, method = 'pearson', tail = 'upper').

## Statistical analyses

Clinical and sociodemographic characteristics, MTBC genotype as well as human genetic ancestry were summarized by the different disease severity measures using proportions or means and standard deviations. Similarly, the human genetic ancestries were summarized by MTBC genotype.

To estimate the associations between disease severity and the explanatory variables bacterial genotype (binary, belonging to Introduction 10 or not) and human ancestries (seBS, eBS, and wBS), we used a logistic regression model. We included three variables as proxies for disease severity: Lung damage based on X-ray score (mild versus severe), TB-score (mild versus severe), bacterial load (continuous, $log_{10}$ transformed). Categorization or transformation of the disease severity measures was performed because the distributions of the residues were not normal, violating the assumption of linear regression. We tested only for Introduction 10 because there were few observations of the other MTBC genotypes. To account for the compositional nature of the human ancestries (i.e. that they sum up to 1), we used the additive log transformation from the R package 'compositions' (*van den Boogaart et al., 2022*). The ancestry proportions were transformed and categorized with category 1 comprising the lowest amount of the respective ancestry and category 3 (4 in the case of wBS) the highest. The categories were chosen to have roughly equal numbers of patients in each. We used categories to allow a non-linear relationship without specifying polynomials and avoid having difficulty in interpretation, but we recognize that a small amount of information is lost. Similar results were also obtained with other parameterizations. We assessed whether there was an interaction between ancestry and genotype on TB severity using the likelihood ratio test. For that, we compared a model including the interaction between the MTBC genotype and human genetic ancestry to a model without the interaction using the 'lmtest' package (*Zeileis and Hothorn, 2002*). The estimates were adjusted for age, sex, smoking, the number of weeks with cough, the socioeconomic status (ratio between household income and number of household members), education (no, primary, secondary, university), drug resistance profile, TB category (new infection or relapse/reinfection), and malnutrition. Only HIV-negative patients were included in the analysis. All statistical analyses were carried out in R (version 4.1.2). Code for the statistical analysis can be found (https://github.com/mzwyer/TB-Dar_Mtb, copy archived at *Zwyer, 2026*).

To conduct the GWAS, we used logistic regression (TB-score and X-ray score) and a linear model (Ct-value) implemented by PLINK (version 1.9). A total of 6,665,541 common (MAF >0.05) human genetic variants were included. Genetic PCs were calculated using GCTA (version 1.93.2). As covariates, we included the top three human genetic PCs to correct for population stratification, along with sex, HIV status, age, cough duration, smoking, socioeconomic status, TB category, malnutrition, education, and drug resistance profile. Genomic inflation factors were below 1 of all tested outcomes (*Figure 3—figure supplement 1*).

## Acknowledgements

Calculations were performed at the sciCORE (http://scicore.unibas.ch/) scientific computing core facility at University of Basel. We thank all TB-DAR staff and study participants. Swiss National Science Foundation (Grant Nos. CRSII5_177163, CRSII5_213514, 10000213, and 10001893) and the European Research Council (Grant No. 883582). The funders had no role in the study design, data collection, and analysis, decision to publish, or preparation of the manuscript.

# Additional information

## Funding

| Funder | Grant reference number | Author |
|---|---|---|
| Schweizerischer Nationalfonds zur Förderung der Wissenschaftlichen Forschung | CRSII5_177163 | Klaus Reither Jacques Fellay Sebastien Gagneux |
| Schweizerischer Nationalfonds zur Förderung der Wissenschaftlichen Forschung | CRSII5_213514 | Klaus Reither Jacques Fellay Damien Portevin |
| Schweizerischer Nationalfonds zur Förderung der Wissenschaftlichen Forschung | 10001893 | Sebastien Gagneux |
| European Research Council | 883582 | Sebastien Gagneux |
| Schweizerischer Nationalfonds zur Förderung der Wissenschaftlichen Forschung | 10000213 | Sebastien Gagneux |

The funders had no role in study design, data collection, and interpretation, or the decision to submit the work for publication.

## Author contributions

Michaela Zwyer, Conceptualization, Data curation, Software, Formal analysis, Investigation, Visualization, Methodology, Writing – original draft, Writing – review and editing; Zhi Ming Xu, Amanda Ross, Formal analysis, Writing – review and editing; Jerry Hella, Resources, Methodology, Project administration; Mohamed Sasamalo, Resources, Data curation, Formal analysis, Investigation, Methodology; Maxime Rotival, Formal analysis, Methodology, Writing – review and editing; Hellen Charles Hiza, Resources, Data curation, Formal analysis, Methodology; Liliana K Rutaihwa, Resources; Sonia Borrell, Resources, Project administration; Klaus Reither, Funding acquisition, Investigation, Project administration, Writing – review and editing; Jacques Fellay, Damien Portevin, Conceptualization, Funding acquisition, Investigation, Methodology, Project administration, Writing – review and editing; Lluis Quintana-Murci, Formal analysis, Supervision, Methodology, Writing – review and editing; Sebastien Gagneux, Conceptualization, Supervision, Funding acquisition, Investigation, Methodology, Writing – original draft, Project administration, Writing – review and editing; Daniela Brites, Conceptualization, Resources, Data curation, Formal analysis, Supervision, Funding acquisition, Investigation, Methodology, Writing – original draft, Writing – review and editing

## Author ORCIDs

Michaela Zwyer https://orcid.org/0000-0002-3864-1503
Jacques Fellay https://orcid.org/0000-0002-8240-939X
Damien Portevin https://orcid.org/0000-0003-2949-9557
Sebastien Gagneux https://orcid.org/0000-0001-7783-9048
Daniela Brites https://orcid.org/0000-0002-8090-2287

## Ethics

Ethical approval for the TB-DAR cohort has been obtained from the Ethikkommission Nordwest- und Zentralschweiz (EKNZ UBE-15/42), the Ifakara Health Institute-Institutional Review Board (IHI/IRB/EXT/No. 24-2020) and the National Institute for Medical Research in Tanzania-Medical Research Coordinating Committee (NIMR/HQ/R.8c/Vol.I/1622). A written informed consent has been obtained from every patient who has been recruited into the TB-DAR cohort.

Reviewer #2 (Public review): https://doi.org/10.7554/eLife.103533.3.sa1
Author response https://doi.org/10.7554/eLife.103533.3.sa2

## Additional files

### Supplementary files

Supplementary file 1. The different ethnic groups with at least 10 members in our cohort and the region and broad geographic location of the original area of the ethnic group. The latitude and longitude are given in decimal degrees. In the case of two associated regions, a location close to the border of the two regions was selected.

Supplementary file 2. Phylogenetic markers selected to identify the introductions. The position is based on the reconstructed reference of the ancestor (*Chang et al., 2015*) and the derived base indicates the base present in the respective introduction. Intro 1 refers to Introduction 1 within L2.2.1, Intro 5 to Introduction 5 within L4.3.4, Intro 9 to Introduction 9 within L1.1.2, and Intro 10 to Introduction 10 within L3.1.1.

MDAR checklist

### Data availability

The bacterial WGS data can be found under the bioprojects PRJEB49562 and PRJNA670836 on ENA and the human WGS and genotyping data under EGAS00001005850 and EGAS00001007216, respectively. Clinical data, statistical analysis, and customized Python scripts are available on https://github.com/mzwyer/TB-Dar_Mtb, copy archived at *Zwyer, 2026*.

The following datasets were generated:

| Author(s) | Year | Dataset title | Dataset URL | Database and Identifier |
|---|---|---|---|---|
| Zwyer M, Hella J, Sasamalo M, Reither K, Fellay J, Portevin D, Gagneux S, Brites D | 2022 | Tuberculosis epidemic in Dar es Salaam, Tanzania | https://www.ebi.ac.uk/ena/browser/view/PRJEB49562 | EBI European Nucleotide Archive, PRJEB49562 |
| Zwyer M, Hella J, Sasamalo M, Reither K, Fellay J, Portevin D, Gagneux S, Brites D | 2021 | TB-DAR Whole Genome Sequencing Study | https://ega-archive.org/datasets/EGAD00001008400 | European Genome-phenome Archive, EGAD00001008400 |
| Zwyer M, Hella J, Sasamalo M, Reither K, Fellay J, Portevin D, Gagneux S, Brites D | 2023 | TB-DAR Genotyping Study | https://ega-archive.org/datasets/EGAD00010002507 | European Genome-phenome Archive, EGAD00010002507 |
| Zwyer M, Hella J, Sasamalo M, Reither K, Fellay J, Portevin D, Gagneux S, Brites D | 2023 | TB-DAR Genotyping Study | https://ega-archive.org/studies/EGAS00001007216 | European Genome-phenome Archive, EGAS00001007216 |
| Zwyer M, Hella J, Sasamalo M, Reither K, Fellay J, Portevin D, Gagneux S, Brites D | 2021 | TB-DAR Whole Genome Sequencing Study | https://ega-archive.org/studies/EGAS00001005850 | European Genome-phenome Archive, EGAS00001005850 |

The following previously published dataset was used:

| Author(s) | Year | Dataset title | Dataset URL | Database and Identifier |
|---|---|---|---|---|
| Menardo F, Brites D, Gagneux S | 2021 | Biogeography of MTB lineage 1 and 3 | https://www.ebi.ac.uk/ena/browser/view/PRJNA670836 | EBI European Nucleotide Archive, PRJNA670836 |

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

# Appendix 1

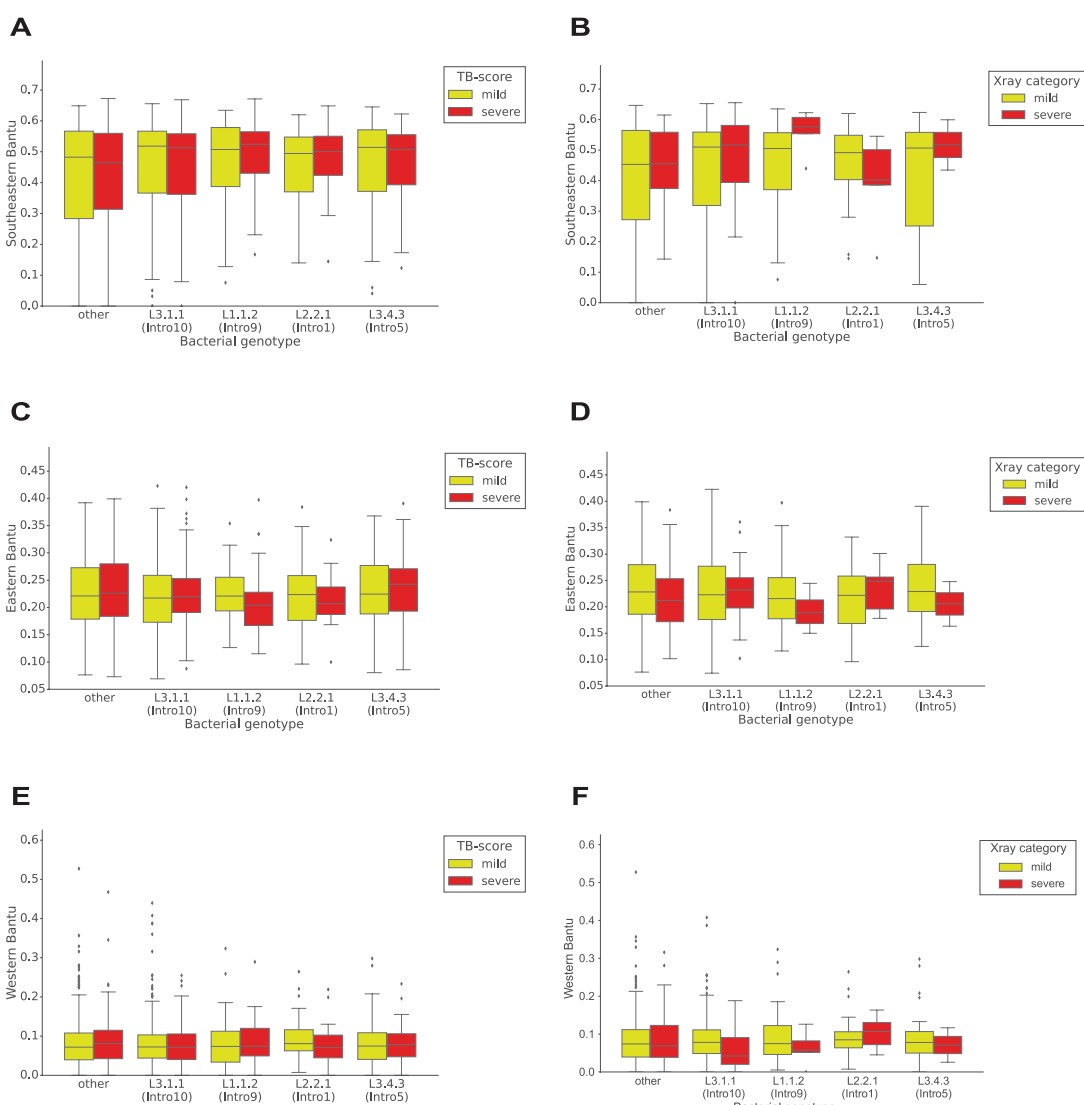

**Appendix 1—figure 1.** Association between the Bantu genetic ancestries and TB-score and X-ray score for each of the most successful introductions. (**A**, **C**, **E**) Southeastern, Eastern, and Western Bantu on mild or severe TB-score, (**B**, **D**, **F**) Southeastern, Eastern, and Western Bantu genetic ancestry lung damage assessed by lung X-ray.

