## [Editor Report · eLife Assessment]

This **valuable** observational study was conducted in Dar es Salaam, Tanzania, to investigate potential associations between genetic variation in *Mycobacterium tuberculosis* and human host vs. disease severity. The authors conclude that human genetic ancestry did not contribute to tuberculosis severity and the evidence supporting this is generally **convincing**. The findings have significance for the understanding of the influence of host/bacillary genetics on tuberculosis disease.

---

## [Referee Report · Reviewer #2 (Public review)]

Summary:

This manuscript reports the results of an observational study conducted in Dar es Salaam, Tanzania, investigating potential associations between genetic variation in *M. tuberculosis* and human host vs. disease severity. The headline finding is that no such associations were found, either for host / bacillary genetics as main effects or for interactions between them.

Strengths:

Strengths of the study include its large size and rigorous approaches to classification of genetic diversity for host and bacillus.

Comments on revisions:

The authors have responded satisfactorily to comments raised.

---

## [Author Response]

The following is the authors’ response to the original reviews.

**Reviewer #1 (Public review):**
Summary:This Tanzanian study focused on the relationship between human genetic ancestry, *Mycobacterium tuberculosis* complex (MTBC) diversity, and tuberculosis (TB) disease severity. The authors analyzed the genetic ancestry of 1,444 TB patients and genotyped the corresponding MTBC strains isolated from the same individuals. They found that the study participants predominantly possess Bantu-speaking genetic ancestry, with minimal European and Asian ancestry. The MTBC strains identified were diverse and largely resulted from introductions from South or Central Asia. Unfortunately, no associations were identified between human genetic ancestry, the MTBC strains, or TB severity. The authors suggest that social and environmental factors are more likely to contribute to TB severity in this setting.Strengths:In comparison to other studies investigating the role of human genetics in TB phenotypes, this study is relatively large, with more than 1,400 participants.The matched human-MTBC strain collection is valuable and offers the opportunity to address questions about human-bacterium co-evolution.Weaknesses:Although the authors had genome-wide genotyping and whole genome sequencing data, they only compared the associations between human ancestry and MTBC strains. Given the large sample size, they had the opportunity to conduct a genome-wide association study similar to that of Muller et al. (https://doi.org/10.1016/j.ygeno.2021.04.024).

Thank you very much for taking the time to carefully review our manuscript and for your suggestions and comments. In another published study using the same cohort (https://doi.org/10.1101/2023.05.11.23289848), we performed a genome-wide association analysis between the genome-wide SNPS of the host and the genome-wide SNPs from the paired MTBC strains. In the current work we were interested in testing specifically if host ancestry and pathogen genotype family, as well as their interaction, were associated with differences in disease severity, a clinical phenotype with direct consequences for both host and pathogen fitness. The study of Müller et al, referred to by the reviewer, investigates whether MTBC families of strains causing disease in two patient cohorts (South Africa and Ghana) were associated with particular human SNPS assessed genome-wide. In that study, clinical phenotypes were not assessed and human ancestries, in a much broader sense than the ones used in our current study, were used as covariates. To leverage the genome-wide information and the clinical variables collected in our study, we have now added a genome-wide association analysis of all the human SNPs with disease severity measures while adjusting for co-variates (age, sex, smoking, cough duration, socioeconomic status, history of previous TB, malnutrition, education level, and drug resistance status) and for human population stratification . Yet, no significant statistical associations were detected (L243-249).

The authors tested whether human genetic ancestry is associated with TB severity. However, the basis for this hypothesis is unclear. The studies cited as examples all focused on progression to active TB (from a latent infection state), which should not be conflated with disease severity. It is difficult to ascertain whether the role of genetic ancestry in disease severity would be detectable through this study design, as some participants might simply have been sicker for longer before being diagnosed (despite the inquiry about cough duration). This delay in diagnosis would not be influenced solely by human genetics, which is the conclusion of the study.

Evidence that mortality and natural recovery from TB vary by disease presentation spectrum come from studies carried out before the introduction of anti-TB chemotherapy. Patients with mild disease presentation, as measured by radiology at the time of diagnosis had higher odds of recovering naturally compared to those with advanced disease (doi: 10.5588/ijtld.23.0254, doi: 10.1164/arrd.1960.81.6.839). Given the deleterious effects of an MTBC infection leading to symptomatic disease on human fitness, we hypothesized that natural selection has acted on human traits underlying TB disease severity. If those traits are heritable one would expect to find underlying genetic variation in human populations. In addition, because certain MTBC genotype families and human populations have co-existed since a least a few centuries to a few millennia, we hypothesized that some of that genetic variation could be related to human ancestry. We have added more details to the introduction to make our rational clearer (L118-127). In our patient cohort, we observed a large variation in disease severity using as approximations; TB-Score, X-Ray score and bacterial burden in sputa (Ct-value as determined with GeneXpert). However, the reviewer is absolutely correct in that patients in our study are being diagnosed at different stages of disease confounding our analysis. This is a limitation of our study which cannot be fully accounted for by including cough duration, as we also acknowledged in the manuscript (L343-346).

Additionally, the study only included participants who attended the TB clinic.

Yes, this is related to the previous point, our study only considers patients that felt ill enough to visit the TB clinic potentially not including patients that had less severe disease as acknowledged.

Including healthy controls from the general population would have provided an interesting comparison to see if ancestry proportions differ.

We agree that it would be interesting to compare the ancestries of healthy controls to the ancestries of TB patients from the same population. However, that would be especially informative with respect to TB susceptibility and would not necessarily be informing disease severity traits and its underlying genetics. The similarities between the ancestry proportions of our cohort with those of neighboring countries such as Kenya, Malawi and Mozambique publicly available genomic data, suggests that there would be no major differences between TB patients and healthy controls.

Although the authors suggest that social and environmental factors contribute to TB severity, only age, smoking, and HIV status were characterised in the study.

Based on the comments of both reviewers, we added the following additional variables as covariates in the regression models: the socioeconomic status representing the ratio between the household income and the number of individuals in the household, malnutrition, the education level and whether it was a relapse/reinfection or a new case.

**Reviewer #2 (Public review):**
Summary:This manuscript reports the results of an observational study conducted in Dar es Salaam, Tanzania, investigating potential associations between genetic variation in *M. tuberculosis* and human host vs. disease severity. The headline finding is that no such associations were found, either for host / bacillary genetics as main effects or for interactions between them.Strengths:Strengths of the study include its large size and rigorous approaches to classification of genetic diversity for host and bacillus.Weaknesses:(1) There are some limitations of the disease severity read-outs employed: X-ray scores and Xpert cycle thresholds from sputum analysis can only take account of pulmonary disease. CXR is an insensitive approach to assessing 'lung damage', especially when converted to a binary measure. What was the basis for selection of Ralph score of 71 to dichotomise patients? If outcome measures were analysed as continuous variables, would this have been more sensitive in capturing associations of interest?

Thank you very much for taking the time to carefully review our manuscript and for your suggestions and comments.

We recruited active TB patients with pulmonary TB disease that were sputum smear-positive and GeneXpert-positive. In this study we aimed at obtaining paired samples from both the patient and the strain, and in the current analysis we aimed at testing if human ancestry and its interaction with the strain genotype could explain differences in disease severity. It is often difficult to obtain microbiological cultures from extra-pulmonary cases and including those cases would have not been possible at the scale of this cohort. We believe as well that extra-pulmonary TB is of less relevance for the question we are addressing because in exclusively extrapulmonary cases, disease severity is not linked with bacterial transmission. However, extra-pulmonary TB can be extremely severe, and it would be very interesting to explore the potential role of human genetic variation underlying extra-pulmonary TB in future studies.

As to the insensitivity of CXR to measure lung damage, we would argue that it depends on what is being assed. As a rationale for the Ralph score, its inventors argue that as in other grading methods, the proportion of affected lung and or cavitation is important to assess severity. It has been described as a “validated method for grading CXR severity in adults with smear-positive pulmonary TB that correlates with baseline clinical and microbiological severity and response to treatment, and is suitable for use in clinical trials” (https://thorax.bmj.com/content/thoraxjnl/65/10/863.full.pdf). While the validation of the score is convincing in that study, and the score has been used in several TB studies and trials, the low proportion of HIV co-infections might have been a limitation. Indeed, as shown in our previous publication, in our cohort of patients, chest X-ray scores were significantly lower in HIV infected TB patients https://doi.org/10.1371/journal.ppat.1010893. In the current analysis, regression analyses performed for the CXR severity and for the other severity measures did not include HIV co-infected patients.

We obtained the same pattern of results using a continuous outcome. However, an assumption of linear regression was violated. The residuals were not normally distributed stemming from the bimodal distribution of the scores in our dataset. The threshold of 71 for the Ralph score has been used by others in previous studies; in its original description it has been suggested as the optimal cut-off point for predicting a positive sputum smear status after two months, which in turn has been shown to predict unfavorable outcomes (https://doi.org/10.1136/thx.2010.136242). Another study showed that a Ralph score higher than 71 was significantly associated with a longer duration of symptoms, higher clinical scores and a lower BMI (doi: 10.5603/ARM.2018.0032).

(2) There is quite a lot of missing data, especially for TB scores - could this have introduced bias? This issue should be mentioned in the discussion.

While we have a TB-score available for each patient, the chest X-ray score is missing for many patients. However, this is random and due both to the absence of an X-ray picture or to the bad quality of X-ray pictures that the radiologists could not assess. When stating that there is a lot of missing data for the TB scores, we assume that the reviewer was referring to the “missing N” columns in Table 1. There, the number of observations missing in each of the disease severity measures actually relates to the explanatory variables (i.e MTBC genotype and human ancestries). This table includes all patients that either had a bacterial genome available or a human genome/genotype (N = 1904). As an example for the TB-score as outcome variable, for 1471 patients the MTBC genotype was determined while it was missing for 433 patients. On the other hand for X-ray scores, 177 had a severe X-ray score, 849 a mild one and for 878 patients, there was no X-ray score available. As for the Ct-value, despite the fact that the patients were recruited based on positive GeneXpert by the clinical team, these results were not always available to us.

(3) The analysis adjusted for age, sex, HIV status, age, smoking and cough duration - but not for socio-economic status. This will likely be a major determinant of disease severity. Was adjustment made for previous TB (i.e. new vs repeat episode) and drug-sensitivity of the isolate? Cough duration will effectively be a correlate/consequence of more severe disease - thus likely highly collinear with disease severity read-outs - not a true confounder. How does removal of this variable from the model affect results? Data on socioeconomic status should be added to models, or if not possible then lack of such data should be noted as a limitation.

Out of the 1904 patients that have either human or bacterial genomic data available, 48 were relapses (2.5%). The mean of the disease severity measures suggest that relapses have a higher CXR score but the TB-score and Ct-values did not differ. Based on the comments of both reviewers, we added the following additional variables as covariates to the regression models: the socioeconomic status representing the ratio between the household income and the number of individuals in the household, malnutrition examined by a doctor, the education level, and whether it was a relapse/reinfection or a new case and if the causative strain had any resistance to any anti-TB drugs. The results did not change. Cough duration could also be a consequence of more severe disease, as pointed out by the reviewer. We present now the results excluding cough duration as a variable from the model, however this also did not affect the results.

(4) Recruitment at hospitals may have led to selection bias due to exclusion of less severe, community cases. The authors already acknowledge this limitation in the Discussion however.(5) Introduction: References refer to disease susceptibility, but the authors should also consider the influences of host/pathogen genetics on host response - both in vitro (PMIDs 11237411, 15322056) and in vivo (PMID 23853590). The last of these studies encompassed a broader range of ethnic variation than the current study, and showed associations between host ancestry and immune response - null results from the current study may reflect the relative genetic homogeneity of the population studied.

We thank the reviewer for these suggestions which we have added to the introduction.

**Reviewer #1 (Recommendations for the authors):**
Minor Comments:(1) The authors should be careful when using the term "Bantu" as opposed to "Bantu-speaking". (i.e. referring to the language group). The term is considered offensive in some settings.

We thanks the reviewer for this important concern, we have revised throughout the manuscript.

(2) There are several "(Error! Reference source not found)" phrases in the place of references throughout the document.

We thank the reviewer for pointing this out, this has been corrected in the revised version.

(3) Please correct line 365: "... sequencing (WGS) the patient...." to "... sequencing (WGS) of the patient...."(4) The figures in the supplementary PDF are not numbered and some are cut-off (I think it is Supplementary Figure S2).

This has been corrected in the revised version.

**Reviewer #2 (Recommendations for the authors):**
Typographical errors(1) There are multiple instances where references have not pulled through to the text, e.g. line 126 (Error! Reference source not found.)

We thank the reviewer for pointing this out, this has been corrected in the revised version.

(2) Line 239: have been show - have been shown?

Thank you, this mistake has been corrected in the revised version.